# Block Copolymer Adsorption on the Surface of Multi-Walled Carbon Nanotubes for Dispersion in *N*,*N* Dimethyl Formamide

**DOI:** 10.3390/nano13050838

**Published:** 2023-02-23

**Authors:** Irena Levin, Aurel Radulescu, Lucy Liberman, Yachin Cohen

**Affiliations:** 1Department of Chemical Engineering, Technion-Israel Institute of Technology, Haifa 3200003, Israel; 2Forschungszentrum Jülich GmbH, Jülich Centre for Neutron Science (JCNS-4) at Heinz Maier-Leibnitz Zentrum (MLZ), D-85747 Garching, Germany

**Keywords:** carbon nanotubes, block copolymer, interfacial adsorption, nanotube dispersion

## Abstract

This research aims to characterize the adsorption morphology of block copolymer dispersants of the styrene-block-4-vinylpyridine family (S4VP) on the surface of multi-walled carbon nanotubes (MWCNT) in a polar organic solvent, *N*,*N*-dimethyl formamide (DMF). Good, unagglomerated dispersion is important in several applications such as fabricating CNT nanocomposites in a polymer film for electronic or optical devices. Small-angle neutron scattering (SANS) measurements, using the contrast variation (CV) method, are used to evaluate the density and extension of the polymer chains adsorbed on the nanotube surface, which can yield insight into the means of successful dispersion. The results show that the block copolymers adsorb onto the MWCNT surface as a continuous coverage of low polymer concentration. Poly(styrene) (PS) blocks adsorb more tightly, forming a 20 Å layer containing about 6 wt.% PS, whereas poly(4-vinylpyridine) (P4VP) blocks emanate into the solvent, forming a thicker shell (totaling 110 Å in radius) but of very dilute (<1 wt.%) polymer concentration. This indicates strong chain extension. Increasing the PS molecular weight increases the thickness of the adsorbed layer but decreases the overall polymer concentration within it. These results are relevant for the ability of dispersed CNTs to form a strong interface with matrix polymers in composites, due to the extension of the 4VP chains allowing for entanglement with matrix chains. The sparse polymer coverage of the CNT surface may provide sufficient space to form CNT-CNT contacts in processed films and composites, which are important for electrical or thermal conductivity.

## 1. Introduction

Carbon nanotubes (CNTs), both single-walled (SWCNTs) and multi-walled (MWCNTs), are of tremendous interest, due to their outstanding properties, attractive for a variety of usage modes and applications [1], such as nanocomposites [2,3], fibers [4], microelectronics [5,6], in particular sensors [7,8,9,10,11] and thermal management [12], energy conversion [13] and storage [14,15], field emitters [16], water treatment [17,18], biomedicine [19,20,21], and more. To benefit from their unique properties, many of those applications require CNTs to be first effectively dispersed in a liquid and integrated into a polymeric matrix that can be further utilized as a finalized product. The quality of the CNT dispersion holds a key role in controlling the desired properties of the resulting composites [22,23]. However, due to the strong vdW attraction forces between individual CNTs [24] and their high aspect ratio, pristine CNTs are often in the form of bundled and entangled agglomerates, which leads to poor dispersibility, both in aqueous and organic solvents, prohibiting their homogeneous dispersion in the polymer matrix [25]. Notwithstanding, both single [26,27] and multiwalled [27] short thin pristine nanotubes, were shown to be amphiphatic. This property enables the formation of stable emulsions in some mixed aqueous/organic solvent systems, whereby the CNTs encapsulate the emulsion microparticles. The emulsions are amenable to further processing without additional dispersants [26,27]. Amphiphilicity was shown to be due to open ends and defects, especially oxidized moieties, providing hydrophilic regions and graphitic surfaces constitute the hydrophobic regions [27]. Obtaining a fine and stable dispersion of individually dispersed CNTs, in particular long unmodified ones, is yet a major practical problem [28].

Amphiphilic block copolymers are successfully used for modifying the solution behavior of CNTs [29]. The difference in solvent selectivity towards the polymer blocks combined with specific interactions between units of the adsorbed block and the CNT surface are the controlling factors in stabilizing CNT dispersions [30]. The adsorbed copolymer blocks serve as an anchor, and the solvophilic blocks extend into the solution, providing stability by steric repulsions between the covered surfaces, or by electrostatic repulsion in the case of polyelectrolyte chains in an aqueous medium [31,32]. The use of amphiphilic block copolymers as dispersants is advantageous when the CNT dispersions are used for the fabrication of composite materials, as the integrity of the CNT/polymer matrix interface that is eventually formed can be improved by entanglement of the dispersing and matrix chains [33]. The process of the self-assembly of Pluronic® block copolymers of poly(ethylene oxide)-b-poly(propylene oxide)-b-poly(ethylene oxide), (PEO-PPO-PEO) has been thoroughly studied in aqueous media. For instance, Shvartzman-Cohen et al. [34] presented modeling of the steric barrier formed by end-attached PEO layers and showed that the strength and range of the steric repulsion induced by the polymers are monotonic increasing functions of both the chain length and polymer surface coverage. Nativ-Roth et al. [31] used molecular theory calculations to show that a threshold Pluronic® concentration is necessary for the formation of stable CNT dispersions and that it depends on the length of each block, the solvent quality, and the type of dispersed CNT (MWCNT or SWCNT).

The quality of CNT dispersion strongly depends on the interaction between the dispersing molecules with the CNTs. Therefore, understanding the nature of this interaction and the relevant parameters, such as the morphology of the dispersant on the nanotube surface, are of great interest. In the case of water soluble linear homopolymers such as poly(vinylpyrrolidone) (PVP) and poly(styrenesulfonate) (PSS) [35], conjugated polymers, such as poly(m-phenylenevinylene-co-2,5-dioctoxy-p-phenylenevinylene) (PmPV) [36] and random coil type biopolymers, such as single-stranded DNA [37,38] and peptides [39], the mechanism of stabilization was suggested to be based on the polymer wrapping around the nanotubes. It has been suggested that to minimize strain in their conformations, some polymers can wrap around nanotubes in a helical fashion, forming a CNT–polymer supramolecular complex [31,35,40]. The ability of the globular protein bovine serum albumin (BSA) to disperse both SWCNTs and MWCNTs in water was studied by Regev and coworkers [41,42]. They concluded that successful dispersion is achieved without significant change in the globular protein conformation [41]. Furthermore, a dynamic equilibrium between the dissolved BSA and a small amount of adsorbed protein (a few %) is required for proper dispersion [42]. In the case of amphiphilic molecules, such as block copolymers, the suggested interaction was described by a “non-wrapping” or “loose adsorption” model, having nonspecific polymer–CNT interactions, which are restricted to the adsorbing block (or end group), and thus do not intervene with the electronic structure of the CNTs or modify their physical properties [31]. This model has been developed on the basis of small angle neutron scattering studies (SANS) of CNT dispersions. Zou et al. [43] reported that the conjugated block copolymer poly(3-hexylthiophene-b-polystyrene) (P3HT-b-PS) self-assembles on the nanotube surface to form a very thin layer, exhibiting a random morphology, without any clear structural pattern. Mountrichas et al. [44] reported on the existence of hemi-micellar structures of the adsorbed amphiphilic block copolymer poly(styrene-b-sodium sulfamate/carboxylate polyisoprene) on the surface of CNTs. Périé et al. [45] dispersed MWCNTs by ABC terpolymer, poly(styrene-b-butadiene-b-methyl methacrylate) in a selective solvent and observed a cylindrical adsorption pattern on the surface of CNTs. Korayem et al. [29] investigated the adsorption morphologies of a commercial dispersant copolymer (BYK 9076) on the surface of MWCNTs and reported the existence of three different morphologies, depending on the copolymer/CNT ratio. Tsarfati et al. studied the dispersion of SWCNTs by amphiphilic polymers composed of polyethylene glycol (PEG) and perylene diimide (PdiI) sequences in water and chloroform. Helical PdiI assemblies adsorbed onto the nanotube surface due to hydrophobic and π–π interactions were observed in aqueous dispersions, whereas a single adsorbed layer of PdiIs was deemed sufficient to disperse individual SWCNTs in chloroform [46].

Small angle scattering techniques are powerful tools for the investigation of the structural properties of colloidal systems. By fitting theoretical models that relate the shape of the scattering patterns predicted by model calculations to the experimental data, one can extract the main structural parameters that are characteristic of the investigated system. The analysis of the resulting scattering pattern provides information about the size, shape, arrangement, and interactions of the components of the sample. Dror et al. [47] investigated SWCNT dispersions stabilized with gum arabic and with a copolymer of styrene and sodium maleate using SANS. The scattering data could not be fitted to the “−1” power law related to the tight wrapping model. Consequently, Dror introduced a “core-chains” model describing the formation of individual and small bundles of SWNTs decorated by a corona of solvent-swollen polymer coils loosely adsorbed to the SWNT surface. This model was consistent with the SANS data in the case of both polymers, although differing in their chemical structure. Granite et al. [48,49] studied the stabilization of SWNTs by amphiphilic triblock copolymers PEO-PPO-PEO in aqueous solutions at different H_2_O/D_2_O ratios using the contrast variation technique. Granite applied Dror’s “core-shell” model [47] to describe the conformation of the Pluronic® F127 copolymer on the surface of SWCNTs and suggested a second and more detailed “core-shell-chains” model which differentiates between the scattering length densities (SLDs) of the different blocks in the block copolymer (PEO and PPO) [48,49]. Golosova et al. [50] performed contrast variation studies on dispersions of covalently modified CNTs (both MWCNTs and SWCNTs) with PS in mixtures of toluene and deuterated toluene solutions. The structures of CNTs and their aggregates were modeled as fractal aggregates of homogeneous cylinders (for SWCNTs) or of core–shell cylinders (for MWCNTs). Kastrisianaki-Guyton et al. [51] repeated the contrast variation experiments performed by Granite et al. [48,49] with Pluronic® F127 and showed that the analysis of polymer adsorption onto SWCNTs can be simplified by using a relatively simple core–shell cylinder model to characterize the adsorbed layer. Though using a different model, their conclusion was consistent with that of Granite et al. [48,49], in the understanding of the adsorbed chains as emanating into the solvent away from the CNT surface so that they can be envisioned as forming a shell with a very high water content around the CNT core. Han et al. [52] presented the tunable encapsulation structure of Pluronic® F127-coated SWCNTs in an aqueous solution. The structure of the hydrated block copolymer encapsulation layer surrounding the SWCNT can be controlled either reversibly by varying the temperature, leading to a transition of F127 layers from loosely attached polymers to more compact individual polymer blobs, or irreversibly by adding 5-methylsalicylic acid (5MS), leading to a significant structural change in the block copolymer layer, from loose corona shell to a tightly encapsulating compact shell. Tehrani et al. [53] studied structural features in SWCNT–PEDOT:PSS nanocomposites using SANS and USANS. In another study, the dispersion grade of phenol-functionalized MWCNTs in polar organic solvents was investigated by SANS [54]. Justice et al. [55] studied composites reinforced with functionalized CNTs via ultra-small angle X-ray scattering (USAXS) and suggested a simplified tube form factor (STFF) to describe the scattering from CNTs. To account for the flexibility of the tube or aggregation, they employed a worm-like chain model, which considers the long-range fractal correlations by implementation of a fractal structure factor.

For many practical applications, noncovalent dispersion of CNTs in organic solvents is essential. In particular, proper dispersion of MWCNTs in an organic solvent is important in fabricating their nanocomposites in a polymer film for electronic or optical devices. Katz et al. [56] successfully used a triblock copolymer of poly(methylmethacrylate)-b-polystyrene-b-poly(methylmethacrylate), (PMMA-b-PS-b-PMMA) to homogeneously disperse MWCNTs in DMF, during the fabrication of aligned composite nanofibers of PMMA for an optical limiting device. Shin et al. [57] reported on the successful dispersion of SWCNTs both in polar and nonpolar solvents by PS-b-P4VP block copolymer. Arras et al. [33] reported on the successful alignment of pristine MWCNTs using high molar mass PS-b-P2VP block copolymer in a nanocomposite film, where PS-b-P2VP acted both as a dispersant in p-xylene and as load transferring matrix during shear-alignment. However, the nanostructure and the interactions between MWCNTs and dispersing polymers in organic solvents have not been studied in detail. Ziv et al. [58] showed that non-fully hydrolyzed poly(vinyl alcohol)—(PVA), containing ~10–20% vinyl acetate (VAc) dissolved in ethanol–water mixtures containing more than 50 vol% ethanol, can successfully disperse MWCNTs, much better than fully hydrolyzed PVA. The importance of swelling the VAc sequences, being the “less solvolphilic” sequences adsorbed onto the CNT surface, was hypothesized, with relevance to the steric (entropic) repulsion dispersion mechanism [58].

This research aims to characterize the adsorption morphology of block copolymer dispersants on the surface of MWCNTs in a polar organic solvent, *N,N*-dimethyl formamide (DMF). More specifically, to examine the interactions between block copolymers and the surface of MWCNTs, when one block, typically the shorter one, has less favorable solvent–polymer interaction and hence is the adsorbed block. The hypothesis is that small-angle neutron scattering (SANS) measurements, using the contrast variation (CV) method, can provide information on the density and extension of the polymer chains adsorbed on the nanotube surface, by quantifying the swelling of both the adsorbed and extended blocks, which can yield insight into the means of successful dispersion.

## 2. Materials and Methods

### 2.1. Materials

The MWCNTs used in this work were obtained from NanoIntegris Technologies, Inc. (Quebec, QC, Canada, catalog no. MW24-170) and were used without additional modification or purification procedures. The nominal characteristics are: outer and inner diameters (OD, ID): <20 and 4 nm, respectively, length 1–12 µm and purity >99%. *N,N*-dimethylformamide (DMF), AR grade, was obtained from Frutarom Ltd. (Haifa, Israel), and was used as received. Fully deuterated DMF (DMF-d7) was purchased from Sigma-Aldrich (Rehovot, Israel). The physical properties of the solvents are listed in Table 1. The block copolymers were purchased from Polymer Source Inc. (Dorval, QC, Canada): S4VP, a block copolymer of poly(styrene-b-4-vinylpyridine) and dS4VP, a block copolymer of poly(deuterated styrene(d8)-b-4-vinyl pyridine. The chemical formulae are shown in Figure 1 and some characteristics are listed in Table 2.

### 2.2. Preparation of MWCNT/Polymer Dispersions and Verification of Constant CNT Content

The MWCNT/polymer dispersions were prepared using the following procedure: The selected copolymer was added to DMF and magnetically stirred for 3 h to form a polymer solution of a desired concentration. This solution was added to MWCNTs powder followed by probe sonication (ultrasonic cell disrupter 2000U, model 51-05-290, Ultrasonic Power Corp., Freeport, IL, USA), equipped with a finger-like tip, at frequency of 20 kHz and capacity of 200 W, for 10 min at ~100 W, to create a dark-colored solution. This solution was then bath sonicated for 60 min in an ultrasonic cleaner bath (DC150H, frequency 40 kHz, MRC Inc., Holon, Israel). Our qualitative experience is that a two-step sonication process resulted in better dispersions containing fewer agglomerates, as evaluated qualitatively by light-microscope imaging. Two-step sonication for MWCNT dispersion has been reported previously [59]. Finally, the mixtures were centrifuged for 20 min at 1900 RCF (CN-2060, MRC Inc., Holon, Israel) and the supernatants were used for the remainder of the process. Dispersions for SANS contrast variation measurements were prepared at different contrasts using mixtures of deuterated and protiated DMF (DMF-d7/DMF). Since the centrifugation process reduces the CNT content in the dispersion by an unknown amount, the following steps were applied to minimize the differences in MWCNTs concentration in the different set of samples. For each sample set, two MWCNT dispersions were prepared by the procedure described above: one using protiated DMF and the other using deuterated DMF-d7 as a solvent. After centrifugation, these two dispersions were used to prepare dispersions at the intermediate contrasts by appropriate mixing. The different sets of samples used in SANS measurements are listed in Table 3.

UV–visible light spectroscopic measurements were carried out to verify that the dispersions in each given set of samples had similar CNT concentrations. The UV–visible absorption spectrum at wavelengths between 200 and 500 nm is sensitive to the CNT concentration [60]. The solution samples and the blank sample (without CNTs) were diluted by a constant factor and absorbance spectra were obtained using a UV-1600PC spectrophotometer (VWR International, Leuven Belgium), at the wavelength range of 200–500 nm, with 10 mL quartz cuvettes. Appendix A in the Appendix A shows an example of the absorbance data of MWCMTs/S_12_4VP_12_ dispersions at different contrasts, demonstrating nearly identical absorption, within experimental error. Similar results were obtained for all sample sets.

### 2.3. Transmission Electron Microscopy (TEM) and Cryogenic TEM (cryo-TEM)

The pristine MWCNTs were examined by TEM imaging, and the quality of their dispersion was evaluated by cryo-TEM, at the Russell Berrie Electron Microscopy Center of Soft Matter, Dept. of Chemical Engineering, Technion. The cryo-TEM methodology, especially as employed at this EM center, has been reviewed [61]. Extension of the cryo-TEM method to non-aqueous systems was also reviewed recently [62]. In particular, cryo-TEM has been used to image CNT dispersions in water [63] and chlorosulfonic acid [64]. Pristine CNTs were dispersed in DMF by rapid mixing and a drop was placed on a TEM grid for solvent evaporation prior to imaging. Preparation of MWCNT/polymer dispersions in DMF for cryo-TEM imaging was carried out in a controlled environment vitrification system (CEVS) [65] to capture the structure in the liquid state. Vitrified samples were prepared in a CEVS at controlled temperature of 25 °C at saturated DMF vapor conditions. The chamber was saturated with the solvent composition of each specific dispersion. A drop of the dispersion (about 3 μL) was placed onto a TEM copper grid covered with perforated carbon film. The sample was immediately blotted 1–2 times in order to minimize the thickness of the resulting liquid film and was rapidly vitrified by dropping it into liquid nitrogen (−196 °C). The specimens were kept inside dewars with liquid nitrogen at cryogenic conditions. The specimen transfer to the microscope was performed using Gatan 626 cryo-specimen holder and transfer station (Gatan Ametek, Pleasanton, CA, USA). Imaging of vitrified dispersions was performed using the FEI Tecnai T12 G^2^ dedicated cryo-TEM (formerly FEI Corp., Hillsboro, OR, USA), equipped with a LaB_6_ emitter, operated at 120 kV. Images were recorded with a high-resolution Gatan UltraScan 1000 CCD camera. Some final images were acquired using a Talos 200C (200 kV) high-resolution TEM (Thermo Fischer Scientific Inc., Waltham, MA, USA, formerly FEI Corp., microscope assembled in Brno, Czech Republic), equipped with a field emission gun electron emitter. Imaging was performed using a Falcon III direct imaging camera for high-resolution imaging at low dose mode and a Volta phase plate for contrast enhancement. Image processing was completed using Gatan Digital Micrograph 2.31.734.0 software package (T12 images) and TIA 5.0 SP4 software (Talos images).

### 2.4. Small Angle Neutron Scattering (SANS)

Dispersions for the contrast variation measurements were prepared as indicated in Section 2.2, at different DMF/DMF-d7 combinations, and the list of the sample sets is given in Table 3. Solutions of the block copolymer without MWCNTs, at the same concentration and contrast, were used as references for the scattering measurements of MWCNTs dispersions (for background subtraction). The SLDs of the solvents in the various DMF/DMF-d7 mixtures are listed in Table 4.

SANS measurements were carried out on the KWS-2 beamline at the FRM-2 research reactor, at the Jülich Center for Neutron Science, Outstation at the Heinz Maier-Leibnitz Zentrum, Garching, Germany [66,67]. The samples were studied in quartz cuvettes of 1 and 2 mm thickness at temperature of 25 °C. The experiments were carried out at 2, 8, and 20 m sample to detector distances at wavelength (λ) of 5 Å, with a wavelength spread Δλ/λ = 20%, and a sample aperture of 8 × 8 mm. Collimation was positioned at the same length as the sample to detector distance (2, 8 and 20 m), resulting in a range of scattering vectors *q* of 0.002–0.345 Å^−1^, where *q* = 4sin(θ)/λ and 2θ is the scattering angle. The data were corrected for the detector sensitivity (using an incoherent plexiglass sample). Electronic noise (using a B_4_C mask) and scattering from an empty cell were subtracted. Intensity calibration on the absolute scale (scattering cross section per unit volume, cm^−1^) was performed using a plexiglass secondary standard. The measured counts from the two-dimensional detector array containing 128 × 128 channels were averaged radially to attain a one-dimensional scattering curve. SANS data were analyzed using the QtiKWS software [68] by Vitaliy Pipitch for data reduction and initial visualization. SANS modeling was carried out by the use of the SasView application [69]. The fitting method employed was nonlinear least squares.

### 2.5. Small Angle X-ray Scattering (SAXS)

Some samples measured by SANS were also measured by SAXS. These measurements were carried out using a Rigaku SAXS/WAXS system (Rigaku Innovative Technologies, Auburn Hills, MI, USA). The system is equipped with a MicroMax-002 + S sealed microfocus tube and two Göbel mirrors, (CuKα radiation λ = 1.542 Å), three pinhole slits, and a generator powered at 45 kV and 0.9 mA. The samples were sealed in thin-walled glass capillaries about 2 mm in diameter and 0.01 mm wall thickness and measured under vacuum at 24 °C. The scattering patterns were recorded by a two-dimensional position-sensitive wire detector (Gabriel) positioned 150 cm behind the sample. The scattering intensity was recorded in the *q* range of 0.0075–0.264 Å^−1^. The scattering from an empty cell and electronic background were subtracted from the measured intensity. The absolute intensity (scattering cross section per unit volume, cm^−1^) was calibrated by normalizing the data with respect to the primary beam intensity using a secondary calibration standard (glassy carbon), transmission, capillary diameter (calculated from its transmission), time, solid angle, and the Thompson factor *T_h._*

## 3. Results

### 3.1. TEM Evaluation of MWCNTs Dimensions

The pristine MWCNTs were visualized by TEM to estimate their dimensions and evaluate the presence of impurities, aggregates, and entanglements. The CNTs were dispersed in the solvent without polymer, thus the dispersion had no long-term stability but was stable enough to prepare samples on the TEM grid. In the TEM images presented in Figure 1, the pristine CNTs exhibit the structure of hollow tubes, well separated and apparently without agglomerates or extraneous impurities. The results of image analysis, based on 100 measurements from different tubes, provide a rough indication of the tubes’ outer and inner diameters as 16.8 ± 1.3 and 4.2 ± 0.7 nm, respectively (the length could not be reliably measured from the TEM images, being mostly longer than the field of view).

### 3.2. Cryo-TEM Imaging of MWCNTs Dispersions in DMF by Adsorbed Copolymers

Cryo-TEM was used to examine the quality of the dispersion in the liquid state and in an attempt to visualize the presence of the dispersant on the surface of the tubes. Low-dose imaging mode was used to minimize the electron beam radiation damage by reducing the number of electrons hitting the sample per unit area. Figure 2 shows cryo-TEM images of a typical dispersion of MWCNTs/S_12_-4VP_12_ in DMF after centrifugation. The images show that the tubes are individually dispersed. Long individual MWCNTs are observed, marked by the two parallel thin dark lines in the gray background of the vitrified DMF. The nanotubes appear mostly intact, with a relatively uniform cross-section dimension. Another observation is that some regions of the tubes seem to be filled with solvent, while others appear to be empty. The latter is evident by the seemingly brighter thin region between the parallel dark lines, due to the empty inner core. The polymer adsorbed on the MWCNT surface, however, could not be imaged due to the low contrast between the polymer and the solvent.

At increased electron dosage, radiation damage by solvent radiolysis is observed, starting as enhanced contrast and appearance of bubble-like features at the interface between the CNT and the vitrified solvent. This is in accord with previous reports of cryo-TEM imaging of CNTs in chlorosulfonic acid (CSA) [64]. Figure 3 presents several images of an MWCNT dispersion by dS4VP, acquired in the same location at an increasing dosage of electron radiation, tracking the evolution of the radiation damage. Comparing the cryo-TEM images of dispersed nanotubes (Figure 2 and Figure 3) with TEM images of pristine nanotubes (Figure 1) indicates qualitatively that the dispersion process did not introduce significant damage. Furthermore, the cryo-TEM images provide evidence of the partial filling of the CNT core with the solvent. The attempt to discern evidence of the adsorbed polymer, whether in low-dose images or by possible enhancement with electron radiolysis was not successful. Although the enhanced radiolysis at the CNT surface might have been considered to indicate the presence of adsorbed radiation-sensitive matter, it cannot be conclusive, as such behavior was observed previously in various nanotubes dispersed in CSA without any additional dispersing agent [64].

### 3.3. SANS Measurements of S4VP Block Copolymer Solutions

Before considering the scattering from the MWCNT dispersions, the state of the block copolymer solutions without CNTs is evaluated, as they serve as background for the SANS measurements of the MWCNT/polymer hybrids. In particular, the issue of possible microphase separation needs to be considered. The scattering patterns from 1 wt.% solution of the S4VP block copolymers of varying PS block molecular weights (MW), 12, 48, and 124 kDa are presented in Figure 4. The measured data are well fit by the Lorentzian function of the Ornstein and Zernike equation, Equation (1), commonly applied to the scattering from semi-dilute homo-polymer solutions [70,71]:(1)$$I\left(q\right)=I\left(0\right){\left(1+\xi^{2}q^{2}\right)}^{-2}+bkg$$
where *ξ* is the screening length, also termed the correlation length, beyond which excluded volume interactions are not relevant, assuming semi-dilute solution behavior [72,73]. *I*(0) is the extrapolated zero angle intensity and *bkg* is the background incorporating the incoherent scattering. The patterns do not exhibit any excess scattering at very small angles above the Lorentzian curve, characteristic of microphase separation in block copolymer solutions, such as observed in a previous study of solutions of PS(73 wt.%)-b-P4VP(37%) (310 kDa) in DMF-d7, at 5 wt.% concentration [74]. Furthermore, no evidence for phase separation could be observed by cryo-TEM imaging, such as reported in that previous study. Thus, the copolymers can be envisioned to be uniformly soluble at the concentration and temperature of this study, and Equation (1) can be considered as a suitable empirical relation for the purpose of background subtraction, bearing in mind that DMF is a good solvent for the homopolymers PS and P4VP [75,76]. As will be discussed further below, the polymer concentration (1 wt.%) is below the overlap concentration for the MWs under study. Nevertheless, Equation (1) is used empirically for background subtraction due to the good fit in the measured *q*-range.

### 3.4. SANS Measurements of MWCNTs Dispersed by the S_12_-4VP_12_ Block Copolymer

The SANS patterns of 0.5 wt.% MWCNT +1 wt.% S_12_-4VP_12_ dispersions at six contrasts (DMF/DMF-d7 mixtures), before and after background subtraction, are shown in Figure 5a,b, respectively. The SANS patterns of the polymer solutions with the Lorentzian fits are presented in Appendix A. The measured scattering patterns cover the range of scattering vectors (*q*) between 0.003 and 0.2 Å^−1^, corresponding to observable length scales of 2π/*q* ≈ 30-2100 Å. Note the increasing incoherent scattering with decreasing content of deuterated DMF in the solvent mixture. Minimal scattering intensity is observed for dispersions in solvent composition between 70% and 80% DMF-d7. The contrast matching point is evaluated at 76% DMF-d7, as shown in the insert in Figure 5b, by plotting the square root of the background-subtracted intensity at a selected low *q* value of 0.0084 Å^−1^, as a function of DMF-d7 content (the data points for 80 and 100% DMF-d7 were taken to be negative). We believe this minimal scattering is the matching point of the MWCNT/S_12_-4VP_12_ hybrid. The scattering patterns at the low *q* range, shown in Figure 5c (*q*~0.002 to 0.009 Å^−1^, equivalent to distances of ~3140–700 Å) exhibit a *I*(*q*)~*q*^−a^ power law dependence with the exponent approximately −1, which is characteristic of thin long rod-like particles (the values range from −1.2 to −1.4), as shown in Figure 5c.

### 3.5. SANS Measurements of MWCNTs Dispersed by the dS4VP Block Copolymer

In order to differentiate between the blocks of the S4VP copolymer adsorbed on the MWCNT surface and those extended towards the solution, the scattering from MWCNT dispersions by a polymer with a deuterated PS block (dS4VP) was compared to that described above from the fully protiated S4VP copolymer of similar block lengths. The scattering patterns of 0.5 wt.% MWCNT + 1 wt.% dS4VP dispersions at five contrasts (DMF-d7/DMF mixtures), before and after background subtraction, are shown in Figure 6a,b, respectively. As described above, copolymer solutions at the same concentration used in preparing the MWCNT dispersions were used as background measurements for the dispersions’ scattering. The data for the dSVP polymer solutions are presented in Appendix A Appendix A and were discussed in the previous section. For the purpose of background subtraction, the data were fit to Lorentzian functions (Equation (1)). In the SANS patterns of the MWCNT/dS4VP hybrids dispersions after background subtraction, shown in Figure 6b, the highest intensity is obtained at 10% DMF-d7 contrast. The lowest scattering intensity is measured at solvent content around 85% DMF-d7. The matching point is evaluated at about 87% DMF-d7, as shown in the inset in Figure 6b. This value is higher than that of the dispersion in the protiated polymer (Figure 5b). It is due to the deuterated PS blocks in the dS4VP copolymer shifting the overall SLD of the hybrid to a higher value, due to the significantly higher scattering length of deuterium compared to hydrogen.

### 3.6. SANS Measurements of MWCNTs Dispersed by S4VP Block Copolymers of Different Styrene Block Molecular Weights

In order to examine the influence of the molecular weight of the block copolymer on the adsorption characteristics and the stability of the CNT dispersions in organic solvents, block copolymers with different block ratios were examined. Two additional S4VP block copolymers were used. All three block copolymers studied have a similar molecular weight of the P4VP block, but they differ in the molecular weight of the PS block, 12, 48, and 124 kDa, as listed in Table 2. The MWCNT/S4VP dispersions with all three copolymers were prepared by the same procedure as described above, and SANS measurements were performed at the same conditions. All three dispersions were found to possess good long-term stability. No significant formation of agglomerates was observed after centrifugation and prolonged incubation for several months. As with the MWCNT/S_12_-4VP dispersion, low-dose cryo-TEM imaging showed well-dispersed, intact tubes, but failed to reveal the presence of the adsorbed polymer on the surface of the nanotubes. SANS measurements were performed with dispersions in DMF solvents at two matching conditions 10% and 100% deuterated DMF-d7, which nearly match the S4VP polymer and the graphitic CNT walls, respectively. Figure 7a,b present the SANS patterns of the three MWCNT/S4VP dispersions with the block copolymers having three different PS block lengths, in the solvent containing 10% DMF-d7, before and after background subtraction, respectively. The patterns of all three hybrid types nearly overlap in the entire measurable *q* range, as expected by the matching of the polymer and solvent SLD. This renders the observed scattering to be due to the MWCNTS only, which are the same in all three dispersions. The variance at very high *q* may be due to issues of background subtraction.

At 100% DMF-d7, the block copolymers are not matched, while the MWCNTs are nearly matched. Figure 8a,b present the SANS patterns of the three MWCNT/S4VP dispersions with the block copolymers having three different PS block lengths, in the solvent containing 100% DMF-d7., before and after background subtraction, respectively. After background subtraction (Figure 8b), it is apparent that the curves overlap in the intermediate and high *q* ranges, which are sensitive to cross-section details of the hollow cylinder of the CNT structure, and the short-range polymer molecular structure, as discussed below. The patterns differ in the lower *q* range (*q* < 0.01 Å^−1^), in which the scattering is due mostly to the elongated MWCNT/polymer hybrid structure. Considering that the MWCNTs are nearly matched at this contrast, the differences in the curves at the low *q* range arise mainly from the differences in the structure of the adsorbed polymers, especially their extension into the solvent, as analyzed in the following sections.

## 4. Discussion

### 4.1. Analysis of S4VP Copolymer Coil Conformations in DMF

Despite the good fits of the Lorentzian function to the measured scattering from the S4VP copolymers at different MWs, shown in Figure 5, Equation (1) is not suitable for the evaluation of the conformation of the block copolymers in solution, as the polymer concentration under study (1 wt.%) is below the overlap concentration (c*) at all the MWs studied. This can be evaluated using the tabulated Mark Houwink parameters (K, a) for the intrinsic viscosity ([η] = KM^a^) of the homopolymers in DMF, PS at 20 °C (0.024 mL/g, 0.63) [77] and poly(2-vinylpyridine)-(P2VP) at 25 °C (0.0147 mL/gr, 0.67) [78]. By the relation c* ~ [η]^−1^ with the proportionality constant about unity, the overlap concentration for a 24 kDa polymer is in the range of 7–8 wt.%, and about 2.5 wt.% for 136 kDa PS. Thus, the copolymer coils in the solution do not overlap. Yet, it is interesting to investigate the state of demixing of the block copolymer coils, due to the repulsive polymer–polymer interactions, as indicated by the Flory interaction parameter (χ_S,4VP_), which is estimated to be between 0.3 and 0.35 [79]. This can be completed using the SANS contrast variation data of the dS4VP copolymer, due to the large SLD difference between the blocks, and measurements at a variety of solvent SLDs prepared by mixing deuterated and protiated DMF. The SANS patterns from 1 wt.% dS4VP solutions in the different DMF solvent mixtures before subtraction of the incoherent scattering background are presented in the Appendix A Appendix A. Only three of the patterns exhibited a noticeable signal above the incoherent background: at solvent compositions containing 10, 85, and 100 wt.% DMF-d7. This is understandable as the calculated zero-contrast solvent composition for this dS4VP copolymer is between 50 and 70 wt.% DMF-d7. The measured patterns after background subtraction are presented in Figure 9. The radii of gyration of each block’s coil, as well as the separation between their centers of mass, can be evaluated by analysis of the apparent radius of gyration obtained from the data at each contrast, by the method developed by Ionescu et al. [80,81] as reviewed by Richards [82]. The apparent mean-squared radius of gyration ($\langle R_{g}^{2}\rangle {}_{app}$), is obtained by fitting the Guinier–Debye function for a random coil conformation, Equation (2) [83,84] to the small angle part of the scattering patterns, for which the excluded volume effect is not yet prominent.
(2)$$I\left(q\right)=I\left(0\right)\left[\exp\left(-\frac{1}{3}q^{2}\langle R_{g}^{2}\rangle {}_{app}+q^{2}\langle R_{g}^{2}\rangle {}_{app}\right)-1\right]/{\left[q^{2}\langle R_{g}^{2}\rangle {}_{app}\right]}^{2}$$

The fitted curves of Equation (2) at low *q* are also plotted in Figure 9. The actual mean squared radii of gyration of each polymeric block, $\langle R_{g}^{2}\rangle {}_{PS}$ and $\langle R_{g}^{2}\rangle {}_{P4VP}$, as well as the mean squared separation between the two coils, $\langle L_{o}^{2}\rangle$, can be obtained from the measurements at the three solvent contrasts by a solution of the three simultaneous equations [82]:(3)$$\langle R_{g}^{2}\rangle {}_{app}=Y\langle R_{g}^{2}\rangle {}_{PS}+\left(1-Y\right)\langle R_{g}^{2}\rangle {}_{P4VP}+Y\left(1-Y\right)\;\langle L_{o}^{2}\rangle$$

In which $Y=w_{PS}K_{PS}/K_{c}$ where $w_{PS}$ is the weight fraction of the PS block. $K_{PS}$ and $K_{c}$ are contrast factors of the PS block and the entire copolymer, respectively, defined as the difference between their scattering length densities and that of the solvent, at each DMF-d7 composition. The fitted parameters of Equation (2) and the parameters of the three Equation (2) at the three contrasts are given in Appendix A Table 1. The solution of Equation (3) yields: $\langle R_{g}^{2}\rangle {}_{PS}^{0.5}\approx 32\;Å,$ $\langle R_{g}^{2}\rangle {}_{P4VP}^{0.5}\approx 38.5\;Å$ and $\langle L_{o}^{2}\rangle {}^{0.5}\approx 44\;Å$. The values obtained for the radii of the individual blocks are about as expected for isolated homopolymer chains of the same M_w_, e.g., by the relation for PS [85]: $R_{g}\approx 1.9M_{w}^{0.59}\approx 29\;Å$. Additionally, the value of $\langle L_{o}^{2}\rangle {}^{0.5}$ is smaller than expected by exclusion of the coiled blocks, for which [80,82]: $\langle L_{o}^{2}\rangle \approx \langle R_{g}^{2}\rangle {}_{1}+R_{g}^{2}{}_{2}$. This result indicates that the connected PS and P4VP coils are slightly overlapped (the extreme case of full overlap will result in $\langle L_{o}^{2}\rangle =0$), and that the polar P4VP coils are somewhat larger than the PS ones, of similar M_w_, due to somewhat more favorable interactions between the polar polymer and solvent.

### 4.2. Analysis of the MWCNT/S_12_-4VP_12_ hybrid SANS Patterns by the Core–Shell Cylinder Model

The structural parameters that are of interest to be determined by analysis of the SANS patterns from the dispersions of the nanotube/polymer hybrids are: (a) the dimensions of the adsorbed polymer shell, i.e., the radial extent beyond the nanotube surface to which the adsorbed polymers extend; (b) the average polymer concentration in this adsorbed polymer layer, which can be interpreted as the average number of polymer chains per unit length; and (c) to differentiate between the adsorption of the different blocks of the copolymer, i.e., to assess if one block adsorbs preferentially onto the CNT surface and if so to evaluate the parameters (thickness and concentration) of the preferentially adsorbed layer. To quantitatively fit the experimental SANS data, several available models relevant to the adsorption of polymers on the surface of CNTs were considered in the past. As mentioned above, in previous studies we used a model that specifically accounts for the adsorbed polymer conformation [47,48,49], which was based on the cylindrical block copolymer model of Pedersen [86,87]. Subsequently, Kastrisianaki-Guyton et al. [51] showed that a simpler core–shell cylinder model can be used to fit data successfully at the different contrasts and obtain important structural information. This model, which is readily accessed in the available software [69], is therefore employed for the data analysis in this study. The model has the following parameters: length (*L* = 2*H* = 10,000 Å, taken arbitrarily as discussed below), core radius *R_core_*, shell thickness *t*, SLDs of the core, shell, and solvent, $\rho_{core}$, $\rho_{shell}$, $\rho_{sol}$, respectively, and the number of cylinders (MWCNT/polymer hybrids) per unit volume of dispersion, *n*.

The number of fitting parameters can be reduced by fixing the SLDs of the components using calculated values. The calculated SLD value of S_12_-4VP_12_ is 1.264 Å^−2^, matched by a 10% DMF-d7 solvent mixture. SLD values of the various solvent mixtures are listed in Table 4. The calculated SLD value of MWCNTs, based on graphitic layers with a density of 1.9 g/cm^3^, is 6.33 × 10^−6^ Å^−2^, which is matched by 100% DMF-d7 solvent. However, the overall MWCNT SLD depends on the actual number of walls in the MWCNTs and on what is contained within its interior (gas or solvent). Partial filling of MWCNTs with solvent was visualized directly in Cryo-TEM images, as shown in Figure 2 and Figure 3. Whether the tubes are empty or not is of minor importance in the measurements of dispersions in low SLD solvent. The influence of the hole is more pronounced in the scattering from dispersions in a fully deuterated solvent. In this contrast the MWCNT graphitic wall is matched, therefore the signal from it, relative to the solvent, is minimal. If the MWCNT is full of solvent, the whole MWCNT is matched and the measured signal after subtraction is mostly from the polymer. If the nanotube core is empty, a distinct signal from the cylindrical core ensues. The presence of solvent within the CNT core is taken into account by the volume of the hole (i.e., the core radius *R_core_*) and its filling fraction with solvent, *f*.

It is important to state the assumptions of the model and the analysis methodology. The dispersed MWCNT/polymer hybrids are assumed to be individually dispersed, i.e., without agglomerates which are considered to be removed by centrifugation. All nanotubes are assumed to be of the same dimensions. Although monodisperse structures lead to zeros in the scattering pattern at larger angles, it is of prime importance to evaluate the fit at small angles with minimum parameters. The polymer shell is assumed to be of constant SLD, being the average of the polymer and solvent at the shell composition. This under-predicts the intensity at larger angles due to neglect of scattering due to inter-polymer correlations (polymer conformation). However, the small-angle part is not perturbed. Solvent intrusion into the nanotube core is accounted for by fitting a value for the SLD of the inner nanotube core, which is assumed to be constant. Finally, the persistence length of the nanotubes is assumed to be much larger than the inverse of the smallest *q* measured, so that they can be considered rigid straight cylinders. Thus, the length is beyond the measurable data. It is given arbitrarily a value of 10,000 Å (1 µm).

The analysis methodology was to first fit the scattering from the MWCNT/S4VP copolymer hybrids in 10% DMF-d7, which matches the polymer SLD. Thus, the core–shell model applies to the MWCNT, providing data on its outer and inner radii as well as the core’s average filling factor by solvent. These parameters were then used in fitting the core–shell cylinder model to the data at 100% DMF-d7, which matched the MWCNT. This provided information on the external polymer shell (thickness and polymer concentration). The obtained structural parameters were then validated by fitting the measurements obtained at intermediate contrasts using the parameters obtained at 10 and 100% DMF-d7, without any adjustable parameters. In fitting the data at these contrasts, the SLD of the MWCNT, taken in its entirety as the cylinder core, was calculated as an average of the graphitic walls’ SLD and that of the inner core, partially filled with solvent, with the filling fraction evaluated by the previous analysis. Again, the prominent fitting is of the high-intensity part at low angles. Finally, the distinction between the PS and P4VP blocks was achieved using the copolymer with a deuterated PS block.

#### 4.2.1. Fitting of the SANS Data of MWCNT/S_12_-4VP_12_ Dispersions in 10 wt.% DMF-d7 Solvent

The core–shell cylinder model was used to fit the scattering data from the MWCNTs dispersions in 10 wt.% DMF-d7 solvent (after background subtraction) as the solvent SLD in this case nearly matches that of the dispersing polymer. It thus allows evaluation of the hollow structure of the MWCNTs: the shell consisting of several graphene layers with a hollow core filled by a fraction of *f* with solvent. The MWCNTs radii evaluated from TEM image analysis were used as a starting point for adjusting *R_core_* and *t* values. As shown in Figure 10, the core–shell model fits well the experimental data at small angles. At larger angles, oscillations appear, due to the form factor of the circular cylinder cross-section in this simplistic model. These oscillations can be smeared by applying some polydispersity of structural parameters, but that would not be useful for the purpose of this study. Table 5 summarizes the parameters obtained by fitting the data of dispersions in 10% DMF-d7 solvent. In addition to the radii of the nanotube core (hole) and its filling factor with solvent (*f* = 0.7), and the thickness of the graphitic layers shell, the analysis yields the number of dispersed nanotubes per unit volume, from which the weight fraction of CNTs can be evaluated (~0.002), indicating retention of ~40% of the CNTs after centrifugation.

It is worth noting that the data at this specific contrast are consistent with the data obtained from SAXS measurements, as shown in Appendix A Appendix A. In SAXS measurements, there is a lack of significant polymer scattering due to low contrast between the electron densities of the polymer and solvent. Thus, the SAXS pattern is equivalent to the SANS measurement at the 10 wt.% DMF-d7. Appendix A displays the scattering data obtained from SAXS measurement of a dispersion of 0.5 wt.% MWCNT + 1% S_12_-4VP_12_ in DMF as compared to SANS data of a similar dispersion in 10% DMF-d7. Appendix A shows the fitted curve of the core–shell model to the combined SAXS/SANS data, generating similar results as listed in Table 5.

#### 4.2.2. Fitting of the SANS Data of MWCNT/S_12_-4VP_12_ Dispersions in 100 wt.% DMF-d7 Solvent and Validation by the Data at Intermediate Contrasts without Adjustable Parameters

At high solvent SLD, when the solvent was DMF-d7, the major contribution to the observed intensity is from the shell of polymer chains adsorbed to the MWCNT surface, as the SLD of the nanotubes is nearly matched. From this measurement, it is possible to estimate the thickness of the polymeric shell, *t*, and the concentration of the polymer in this shell, which can be calculated from the fitted value for the average SLD of the shell. The cylinder core in the model fitting procedure was taken as the entire MWCNT, with the parameters evaluated from the measurement in the 10 wt.% DMF-d7 solvent. Thus, fixed values in the fitting were the core radius (85 Å, the sum of *R_core_* and *t* in Table 5); the MWCNT SLD (taken as 6.21 × 10^−6^ Å^−2^, accounting for its inner hole filled to a fraction of 0.7 with solvent) and solvent SLD (6.33 × 10^−6^ Å^−2^). Figure 11 shows the fit of the core–shell cylinder model to the SANS pattern of the MWCNT/S_12_-4VP_12_ dispersions in DMF-d7, and the evaluated parameters are presented in Table 6. From this fitting, the thickness of the polymeric shell was evaluated to be 110 Å, which is reasonable considering the estimated diameter of the free P4VP coil evaluated in Section 4.1 (end-to-end distance $\sim \sqrt{6}R_{g}\sim 94\;$Å). The evaluated polymer fraction calculated from the fitted shell SLD (6.29 × 10^−6^ Å^−2^) is 0.85%. Interestingly, this value is close to but smaller than the polymer concentration in the original dispersion. It is about 10% of the estimated overlap concentration for this polymer (assuming both blocks maintain their solution conformation, which will be shown subsequently not to be the case for PS). Thus, the adsorbed polymer chains are not crowded around the MWCNT. Furthermore, from the nanotube dimensions (Table 5) and shell thickness and polymer volume fraction (Table 6), the amount of adsorbed polymer per nanotube length can be estimated as ~2.5 Å^−1^, or ~1.9 g/m^2^.

The data at intermediate contrasts were used to validate the model parameters obtained by fitting the measurements at 10 and 100% DMF-d7 solvent (Table 4 and Table 5). The requirement was that now the data should be consistently fitted without any additional adjustable parameters. Using the relevant SLDs at each contrast, the scattering intensity was calculated using the core–shell cylinder model. The SLD of the MWCNT was calculated, as before, using the dimensions and solvent filling factor determined previously, as tabulated in Appendix A Appendix A. The plots of the data and calculated model patterns are shown in Appendix A Appendix A. The accurate fit between data and model calculation in the most relevant low *q* region without adjustable parameters, shown in Appendix A, attests to the validity of the analysis.

### 4.3. Analysis of the SANS Data of MWCNT/dS_11.5_-4VP_11.3_ Dispersions

It is reasonable to assume that the PS blocks of the copolymers have a better affinity towards the MWCNT surface due to the aromatic structure common to both, whereas the P4VP blocks have better solvent interactions due to their polar structure. In order to distinguish between the PS and P4VP blocks, as the polymer adsorbs onto the MWCNT surface, especially as it may be assumed that the PS blocks are preferentially adsorbed, SANS measurements were performed on MWCNT dispersions with a copolymer of deuterated PS and protiated P4VP, of near-equal molecular weights (11.5 and 11.3 kDa, respectively). Since in this case there is no solvent composition that matches the entire polymer in SANS, SAXS measurements were used to evaluate the number of dispersed CNTs per unit volume (*n*), using the previously determined MWCNT structural parameters (Table 5). Figure 12a shows the SANS data of the 0.5 wt.% MWCNT + 1 wt.% dS_11.5_-4VP_11.3_ dispersion in 10 wt.% DMF-d7 solvent, as well as the SAXS data brought to the same intensity scale, demonstrating the good correspondence of the two data sets. Figure 12b presents the fit to the SAXS data of the core–shell model, using the previously determined MWCNT parameters, with only *n* as the adjustable parameter, yielding a value of 4.5 × 10^12^ cm^−3^, about 6% lower than determined for the dispersions with S_12_-4VP_12_ copolymer. The CNT weight fraction in dispersion, 0.0019, and retention of about 38% of the original amount of MWCNTs dispersed, can be estimated from the fitted value of *n*.

The value of *n* thus obtained was used in fitting the core–shell cylinder model to the SANS data of dispersion in 10% DMF-d7 solvent, which matches the SLD of P4VP, as shown in Figure 13a. In this case, the modeled shell is composed of dPS and solvent surrounding the MWCNT, which is the core in the model (including the solvent content in the MWCNT hole, treating the entire MWCNT as the homogeneous core of averaged SLD). The fitted parameters are thus the thickness (*t_PS_*) and dPS volume fraction (Φ_PS_) in this PS polymer shell surrounding the CNT surface. The obtained values of these parameters are 20 Å and 6%, respectively. Fitting the core–shell cylinder model to the SANS data from the MWCNT dispersion with dS_11.5_-4VP_11.3_ in 100 wt.% DMF-d7 solvent, shown in Figure 13b, allows estimation of the thickness (*t_P4VP_*) and volume fraction (Φ_P4VP_) of the P4VP shell surrounding the dPS shell. In this solvent, the SLD of the dPS shell is very close to that of the MWCNT, so they can be assumed together as the cylinder core. The fitted parameters obtained from the three fitting procedures (SAXS and SANS in 10 and 100% DMF-d7 solvent) are summarized in Table 7. Finally, validation of these parameters is confirmed by the good fit of the data obtained by SANS from dispersions in the intermediate contrasts (50, 70, 85 wt.% DMF-d7), without additional adjustable parameters, as shown in the Appendix A Appendix A. This analysis indicates that PS blocks adsorb more tightly (20 Å layer containing 6% PS), whereas P4VP chains emanate into the solvent forming a thick (110 Å) but dilute (0.85%) layer, indicating chain extension. The correspondence of the thickness and volume fraction obtained from analysis of the measurements using both copolymer types (with deuterated and protiated PS) indicates that the P4VP chains penetrate the PS layer.

### 4.4. Analysis of the SANS Data of MWCNT Dispersions by S4VP Copolymer of Varying PS Block M_w_

To examine the influence of the molecular weight of the block copolymer on the adsorption characteristics and the stability of the CNT dispersions in organic solvents, block copolymers with different block ratios were examined. Two additional S4VP block copolymers were available, in addition to S_12_-4VP_12_, sharing similar P4VP block molecular weight while that of the PS block varies, as presented in Table 2. The calculated SLDs and solvent composition (DMF-d7 content) that matches the polymer SLD are listed in Table 8. MWCNT dispersions were prepared according to the same procedure, and initial MWCNT and polymer contents, as described in Section 2.2 (Table 3). All three dispersions were found to possess good long-term stability, without significant formation of agglomerates after centrifugation and prolonged incubation for several months, as observed visually and by cryo-TEM.

SANS measurements of dispersions with the S_124_-4VP_12_ and S_48_-4VP_11_ were performed only with the 10 and 100 wt.% DMF-d7 solvent composition. At the 10% DMF-d7 contrast, all three block copolymers are nearly matched; therefore, their contribution to the scattering intensity is minimal as shown in Appendix A. In this contrast, the dominant contribution to the scattering intensity from the polymer solutions is due to the incoherent scattering from hydrogen atoms. The scattering from the MWCNTs dominates the patterns from the hybrid dispersions. This is seen in Appendix A, before and after background subtraction, respectively, where the scattering patterns of all three hybrid types almost overlap in the entire *q* range. At 100% DMF-d7, the block copolymers are not matched, while the MWCNTs are nearly matched. The block copolymers differ in the length of the PS blocks. Figure 4 showed that the scattering intensity of the bare polymer increases with the Mw of the PS chain. Figure 14 shows the SANS patterns of MWCNT/S4VP dispersions, with copolymers at different molecular weights, in 100% DMF-d7 solvent: (a) before, and (b) after background subtraction, where a difference in the scattering patterns can be noticed at small angles, due to differences in the thickness of the polymer shells. As before, the core–shell cylinder model was fit to the data at the higher PS molecular weight, using the MWCNT structural and solvent content parameters determined previously. The CNT content in dispersion, *n*, was the only fitted parameter at 10 wt.% DMF-d7 solvent composition, and parameters of the polymer shell (thickness and polymer volume fraction) are the only adjustable parameters at 100% DMF-d7. The parameters obtained by the fitting procedures for dispersions in all polymers are summarized in Table 9. The effect of the molecular weight of the PS blocks, the segments that preferentially adsorb to the CNT surface, can be inferred from this analysis. Increasing the length of the PS blocks results in lower retention of MWCNTs in dispersion after the sonication and centrifugation steps of the dispersion process, indicating that it is not beneficial to utilize long PS blocks in MWCNT dispersion in DMF, and possibly other polar organic solvents. As the length of the PS block increases, so does the thickness of the polymeric shell, whereas the total polymer content in the shell decreases. However, a 10-fold increase in PS chain length increases the shell thickness by a factor of 1.7, smaller than a factor of about 4.7, which is expected from the ratio of gyration radii of free PS chains at these conditions. This is due to the densification of the PS chains around the CNT surface at a concentration significantly higher than that within a freely dissolved PS chain. Unfortunately, similar copolymers with deuterated P4VP blocks were not available for this study, to validate these suggested insights.

## 5. Conclusions

This research aimed to characterize the nanostructure of a diblock copolymer that is very successful in dispersing MWCNTs in polar organic solvents such as DMF, a copolymer of poly(styrene) and poly(4-vinyl pyridine) of modest molecular weight (~12 kDa, each block). The general consideration assumed that the aromatic PS segments preferentially adsorb to the nanotube surface, while the polar 4-vinyl pyridine blocks extend into the solvent due to preferential polymer–solvent interactions. An optimal dispersion procedure was utilized, combining ultrasonication followed by centrifugation, resulting in individually dispersed nanotubes as seen in cryo-TEM images. These images also demonstrated the partial filling of the nanotube hole with solvent. The details of the polymer adsorption were studied by careful SANS measurements with contrast variation, using solvent mixtures of deuterated and protiated DMF, which were analyzed by fitting core–shell cylinder models. The designation of core and shell depended on the contrast conditions. It was verified spectroscopically that each sample in a contrasting series had the same CNT content. The SANS measurements were augmented by SAXS, in which contrast due to the adsorbed polymer is negligible, similar to contrast matching of the polymer SLD in SANS. These measurements provided the average MWCNT structural properties (inner and outer radii and filling factor of the inner hole with solvent). SANS measurements at full contrast (using 100% DMF-d7 as solvent) provided information on the polymer shell extending into the solvent. The use of a copolymer with deuterated PS allowed differentiating between a thinner shell of adsorbed PS surrounding the nanotube, and a thicker albeit more dilute P4VP shell extending farther away from the CNT surface. These results were validated by fitting the data at intermediate contrasts without any additional adjustable parameters.

The evaluated number density of MWCNT/polymer hybrids in the dispersion indicated a retention of about 40% of the initial MWCNT content of 0.5 wt.%. From the quantitative analysis of the scattering data of MWCNTs dispersions, it appears that the block copolymer adsorbs onto the MWCNT surface as a continuous yet sparse coverage. The PS blocks adsorb more tightly (forming a 20 Å layer containing about 6% PS), whereas P4VP chains emanate into the solvent forming a thick (110 Å) but dilute (0.85%) layer, indicating chain extension. These results are relevant for the ability of dispersed CNTs to form a strong interface with matrix polymers in composites, due to the extension of the 4VP chains allowing for entanglement with matrix chains. The sparse polymer coverage of the CNT surface may provide sufficient space to form CNT-CNT contacts in processed films and composites, which are important for electrical or thermal conductivity.

A comparison of three dispersants from the S4VP family having a similar molecular weight of the P4VP block but varying length of the PS block revealed that as the length of the PS block increases, so does the thickness of the polymeric shell, and the fraction of the polymer that is adsorbed on the tube decreases. More importantly, the MWCNT retention in dispersion, after the sonication and centrifugation procedures, is lower at increased PS block length, despite this being the preferentially adsorbed segments. Thus, it may be inferred that S-4VP copolymers are effective dispersants of MWCNTs in DMF, and possibly other polar organic solvents, by virtue of preferential interactions of the PS segments with the nanotube surface on one hand, and preferential solvent interactions of the P4VP segments on the other. Moderate molecular weight (~12 kDa) seemed to provide better dispersion.

## Data Availability

Not applicable.

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
