# Peer review of "Block Copolymer Adsorption on the Surface of Multi-Walled Carbon Nanotubes for Dispersion in N,N Dimethyl Formamide"

_nanomaterials, 2023, doi:10.3390/nano13050838_

Round 1

Reviewer 1 Report

The article corresponds to the subject of the journal and is devoted to the study of adsorption layers arising from the adsorption of S4VP block copolymers with a significantly different molecular weight of the PS fragment on the MWCNT surface. Such studies are important for the development of methods for creating CNT dispersions with controlled properties.

The paper presents a large amount of experimental material on the characterization of the obtained samples by the methods of small-angle neutron scattering. Unfortunately, there are no explicitly formulated conclusions in the article. From the presented data, it is difficult to understand how the variation in the composition of S4VP affects the target properties of the resulting dispersions and what these target properties are. It is necessary to add a section “Conclusion” to the article, in which such conclusions will be formulated.

The following clarifications should be made in the text:

- it is necessary to give a transcript of the abbreviation PS when it is used for the first time;

- it would make sense to explain the reasons for using two-stage ultrasonic treatment in the synthesis of MWCNT/polymer composites;

- what conclusions on the subject of the article can the authors draw on the basis of the microphotographs shown in Figures 1,2,3?

- to what extent can the data obtained in the work evaluate the presence and properties of the CNT-CNT contact mentioned in the abstract?

Reviewer 2 Report

This manuscript focuses on characterization bonding between amphiphilic S4VP copolymer on the surface of multi-walled carbon nanotubes (MWCNT) in DMF, a polar organic solvent. Samples were chaacterized using SAXS, SANS, TEM. Authors found that polymers create tight continuous coverage on MWCNTs when the concentration of polymer is low. With increase of polymer concentration in the solution polymer forms thicker shell but less tight around nanotubes. 

The topic is very interesting however, authors need to address the following comments before this manuscript might be published.

1)Authors claim that it is necessary to use amphiphatic copolymers in order to process CNTs in water solution or water-organic solwent mixtures. However, it is possible to process short and thin MWCNTs in aquetous media without additional copolymers due to their amphiphatic nature - see: 10.1021/la026883k and 10.1002/admi.202202407.

2)What was the quality of used CNTs (level of defects, conceration of oxygen bearing groups)? How many of them where open-end?

3)How the size (thickness and length) of used MWCNTs influence interactions with S4VP copolymer? 

(i)What would happen if amphiphatic nanotubes (10.1002/admi.202202407) thinner (below 10 nm in outer diameter)  and short (1.5 micro m in length) were used? How the interactions between copolmer and nanotubes were impacted?

4)It would be good to perform short MD simulations of MWCNT and copolymer in DMF at low and high concentration of polymer to better understand the differences in forming coverage in those two different situations.

Therefore, I recommend this manuscript for publication in Nanomaterials after major revision.

Reviewer 3 Report

The authors studied the adsorption morphology of block copolymer dispersants of the styrene-block-4-vinylpyridine family on the MWCNT surface in N,N-dimethyl formamide using small-angle neutron scattering measurements. Through the controlled experimental procedure for making polymer-assisted CNT dispersion and proper analysis for a combination of control samples, this research will be helpful to understand the detailed interactions between block copolymers and the surface of MWCNTs, probably leading to good dispersion of CNTs in organic solvents or polymer matrix. There are no special questions or comments because the thesis is highly complete and written in English well. 

Author Response

We thank the reviewer for supportive comments.

Some explanatory sentences were added to the introduction (lines 43-52), experimental (lines 248-251), results (lines 388-397) and conclusions (lines 903-907) sections, to improve the manuscript.

Round 2

Reviewer 1 Report

Necessary explanations and clarifications have been added to the article. In the presented form, the article can be published.

Reviewer 2 Report

I am satisfied with the authors' reply and  supplied corrections and I recommend this paper for publication in Nanomaterials.